# Ideas and perspectives: Allocation of carbon from Net Primary Production in models is inconsistent with observations of the age of respired carbon

Carlos A. Sierra[1,2], Verónika Ceballos-Núñez[3], Henrik Hartmann[1], David Herrera-Ramírez[1], and Holger Metzler[2]

[1]Max Planck Institute for Biogeochemistry, 07745 Jena, Germany
[2]Swedish University of Agricultural Sciences, 75651 Uppsala, Sweden
[3]Leipzig University, 04103 Leipzig, Germany

**Correspondence:** Carlos A. Sierra (csierra@bgc-jena.mpg.de)

**Abstract.** Carbon allocation in vegetation is an important process in the terrestrial carbon cycle; it determines the fate of photo-assimilates and it has an impact on the time carbon spends in the terrestrial biosphere. Although previous studies have highlighted important conceptual issues in the definition and metrics used to assess carbon allocation, very little emphasis has been placed on the distinction between allocation of carbon from gross primary production (GPP) versus allocation from net primary production (NPP). An important number of simulation models and conceptual frameworks are based on the concept that C is allocated from NPP, which implies that C is respired immediately after photosynthetic assimilation. However, empirical work that estimates the age of respired $CO_2$ from vegetation tissue (foliage, stems, roots) shows that it may take from years to decades to respire previously produced photosynthates. The transit time distribution of carbon in vegetation and ecosystems, a metric that provides an estimate of the age of respired carbon, indicates that vegetation pools respire carbon of a wide range of ages, on timescales that are in conflict with the assumption that autotrophic respiration only consumes recently fixed carbon. In this contribution, we attempt to provide compelling evidence based on recent research on the age of respired carbon and the theory of timescales of carbon in ecosystems, with the aim to promote a change in the predominant paradigm implemented in ecosystem models where carbon allocation is based on NPP. In addition, we highlight some implications for understanding and modeling carbon dynamics in terrestrial ecosystems.

## 1 Introduction

Carbon that enters the terrestrial biosphere through photosynthesis may have very different fates depending on where this carbon is allocated in plants (Trumbore, 2006). Most of the organic carbon in the biosphere returns to the atmosphere in the form of $CO_2$ via respiration from autotrophic and heterotrophic organisms. The time it takes for assimilated carbon to return to the atmosphere (i.e., the transit time of carbon) depends strongly on what plant part or chemical compound the carbon is allocated to (Rasmussen et al., 2016; Luo et al., 2017; Lu et al., 2018; Herrera-Ramírez et al., 2020). For example, simple sugars may be used quickly for catabolic activity and appear in the respiration flux only a few hours after their biosynthesis, or they

may be used to build structural compounds that can remain stored as biomass for years to decades (Hartmann and Trumbore, 2016). Some of the biomass can be transferred to the soil as litter or via root exudation where it can stay as soil organic matter for even longer periods of time. During the time carbon is stored in the terrestrial biosphere, it does not contribute to the atmospheric greenhouse effect (Neubauer and Megonigal, 2015; Sierra et al., 2021a); therefore, it is of fundamental importance to study carbon allocation and the time carbon stays in ecosystems to improve our understanding of interactions and feedbacks between the terrestrial biosphere and the climate system.

Despite recent advances in the understanding of physiological-level mechanisms of autotrophic respiration (Ra) and carbon allocation in plants (Hartmann and Trumbore, 2016), the representation of these processes in ecosystem models remains overly simplistic. In some models, autotrophic respiration is represented as a proportion of gross primary production (GPP), in other models it depends on the amount of biomass present (Collalti et al., 2020), and in other models it is represented as a combination of both production and biomass. After autotrophic respiration is calculated, the remaining non-respired carbon (net primary production, NPP) is allocated to different plant parts according to specific partitioning coefficients (Franklin et al., 2012; Ceballos-Núñez et al., 2020). This approach appears pragmatic for modeling ecosystem-level carbon balances because it simplifies the representation of autotrophic respiration as one single integrated flux, without additional complexity required to represent respiratory processes from single pools such as stem and roots. However, we argue here that for a more in depth understanding of the fate of photosynthates and the time carbon stays in ecosystems, Ra and carbon allocation functions need to be revisited in many models so to avoid predictions in conflict with empirical observations.

In individual plants, carbon allocation is a highly dynamic process that changes during plant ontogeny to allow them to respond to changes in the environment. Carbon allocation to individual plant parts and their corresponding respiration is often decoupled from GPP and biomass (Collalti and Prentice, 2019). For example, when plants become carbon limited, as it may happen during environmental stress like drought, cold or defoliation, the proportional provision of carbon to Ra decreases, likely to free up resources to maintain allocation to defense (Huang et al., 2019a, b). Plant parts that are cut off from canopy photosynthate supply (and thus from GPP) via girdling respire carbon that is years to decades old (Muhr et al., 2013), where Ra is then fueled with carbohydrates that are stored in older tissues. During environmental stress, and during release of stress, belowground Ra recovers faster than assimilation (Hagedorn et al., 2016), again highlighting a situation where Ra is decoupled from GPP and biomass.

A more mechanistic representation of Ra and carbon allocation in models would improve predictions of the dynamic response of terrestrial ecosystems to environmental changes. In particular, the source of the carbon (GPP or NPP) used for carbon allocation in models have consequences to predict the timescale of ecosystem responses as we will show here. Consequently, in this manuscript we: (1) review models and conceptual frameworks on the main approaches used to represent Ra and carbon allocation at the ecosystem level; (2) show that models that allocate carbon from NPP and not from GPP predict a transit time equal to zero for the entire autotrophic respiration flux, or in other words, respired carbon from vegetation pools has an age (time since assimilation) equal to zero; (3) demonstrate that this prediction is inconsistent with measurements of the age of respired carbon obtained with radiocarbon measurements and does not capture the variability in the transit time of carbon within vegetation; (4) highlight that the choice of carbon allocation approach has consequences for predicting iso-

topic exchange fluxes with the atmosphere, to predict the transit time distribution of carbon in the terrestrial biosphere, and to incorporate radiocarbon measurements in model-data assimilation.

## 2 Historical context, concepts, and models

### 2.1 Conceptual support for allocating carbon from NPP

A common approach to represent autotrophic respiration (Ra) in ecosystem models is to obtain Ra as a constant proportion of GPP, and subsequently allocate NPP to different vegetation pools. This approach is based on the work of Waring et al. (1998), who found constant proportions between NPP and GPP in forest ecosystems, with a constant ratio NPP/GPP = 0.47, or Ra/GPP = 0.53. These constant ratios promoted a simplification in the representation of production and growth in models, with NPP often computed as 50% of annual GPP. Synthesis studies have challenged the constancy of these ratios for different biomes, stand ages, climates, and soils (DeLucia et al., 2007; Collalti and Prentice, 2019). Although some models may continue using a constant ratio to obtain Ra, many other models have now more dynamic implementations to obtain Ra. Nevertheless, the practice of removing Ra immediately after computing GPP, and subsequently allocating NPP to different plant parts seems to be common to most models.

Although a large proportion ($\sim 50\%$) of assimilated carbon may be respired on an annual basis from ecosystems as postulated by Waring et al. (1998), this carbon is not necessarily fixed from the current year or growing season. Instead, photo-assimilates and structural tissues of different ages contribute to the total respiratory flux as we will see below.

Amthor (2000) identified three main paradigms generally used to conceptualize the process of autotrophic respiration: (1) the growth-and-maintenance-respiration paradigm (GMRP), (2) the growth-and-maintenance-and-wastage-respiration paradigm (GMWRP), (3) and the general paradigm (GP) that recognizes all possible processes that respiration might support.

These paradigms are very important to conceptualize the main processes of plant metabolism involved in respiration, but they are not necessarily explicit about the source of carbon that would contribute to the respiration flux. For instance, one can implement a model that computes Ra following the GMWRP, but the actual carbon used for respiration can be subtracted directly from GPP following Waring et al.'s (1998) approach. Carbon would not enter any plant part, but still it would be respired following some physiological concepts.

Research on the matrix approach (Luo et al., 2017, 2022), which shows that one single equation generalizes the majority of existing ecosystem and land-surface models, suggests that Ra is generally subtracted directly from GPP independently of the respiration paradigm implemented in the model. The matrix representation of Luo et al. (2022) can be written as

$$\frac{\mathrm{d}\boldsymbol{x}}{\mathrm{d}t} = U(t)\,\boldsymbol{b} - \xi(t)\mathbf{A}\,\mathbf{K}\,\boldsymbol{x}, \tag{1}$$

where $\boldsymbol{x}$ is a vector of ecosystem carbon pools, $U(t)$ is a function of carbon inputs to the ecosystem, generally obtained as $U(t) = \mathrm{GPP}(t) - \mathrm{Ra}(t) = \mathrm{NPP}(t)$. Then, NPP is allocated to ecosystem compartments such as foliage, wood, and belowground biomass according to the vector of allocation coefficients $\boldsymbol{b}$. The product of the matrices $\xi(t)$, $\mathbf{A}$, and $\mathbf{K}$, is a

compartmental matrix that has in its main diagonal the rates at which carbon is processed in each of the compartments, and in its off-diagonal the rates of carbon transfer among compartments. For vegetation compartments, 100% of all outputs (from mortality and litterfall) are transferred to litter and soil pools, because autotrophic respiration is already accounted for in the first term of equation (1). This modeling choice implies that the carbon used for autotrophic respiration never enters a particular vegetation compartment and does not spend any time there (Figure 1).

In addition to modeling studies, the concept of quantifying carbon allocation after accounting for autotrophic respiration losses is also used in some empirical studies. For instance, the conceptual framework often used to analyze inventory data in tropical forests (e.g. Malhi et al., 2011, 2015) assumes that biomass growth results from the allocation of the products of NPP, after autotrophic respiration occurs. In this case however, carbon allocation is understood as *partitioning* of total NPP. Litton et al. (2007) showed that carbon allocation can be understood differently by different authors, as a flux, as biomass, or as partitioning of the total GPP flux. In the case of the tropical forest data, carbon allocation is understood as partitioning coefficients of the NPP flux and not partitioning of GPP as originally defined by Litton et al. (2007).

Together, these previous studies show that empirical work has promoted the implementation of Ra as a proportion of GPP, or based on some respiration paradigms, but subtracting Ra from GPP before carbon allocation occurs. Therefore models compute first NPP and subsequently allocate the non-respired carbon to plant parts (Figure 1). Any model that could be written using the matrix equation with $U = \mathrm{NPP}$ (equation 1) would allocate the products of NPP and not GPP, independent of the respiration paradigm described by Amthor (2000).

In the following section, we look with more detail at the structure of some particular models with the aim of exploring the main source of carbon used for respiration and allocation.

## 2.2 Representation of C allocation in models

We reviewed the mathematical structure of 19 ecosystem models, with particular attention to the functions implemented for autotrophic respiration and carbon allocation. We found that eleven of these models calculate a net carbon gain by subtracting both growth and maintenance respiration from GPP before carbon allocation occurs. In this group, maintenance respiration is generally computed based on the stock of carbon or nitrogen in vegetation pools, but it is often the case that the source of the respired carbon is the GPP flux and not the carbon stored. These models include ISAM (Masri et al., 2013), IBIS (Foley et al., 1996), CTEM (Arora and Boer, 2005), HAVANA (Haverd et al., 2016), JeDi-DGVM (Pavlick et al., 2013), and the model proposed by Trugman et al. (2018). In CLM4.5 (Oleson et al., 2013) for example, maintenance respiration depends on temperature and the stock of nitrogen in each structural carbon pool, but the required amount of carbon needed to maintain existing tissue is subtracted from total GPP. In case the respiratory demand is larger than the available C from GPP, a storage pool provides the necessary amount of carbon for maintenance respiration. Growth respiration is computed as a proportion of the new carbon allocated to growth, and occurs immediately after this allocation occurs, i.e. growth respiration is subtracted from new carbon, despite the presence of NSC pools that only support growth (Oleson et al., 2013). In ACONITE (Thomas and Williams, 2014), there is a maintenance respiration compartment that receives C from the labile and bud (a pool that stores C before allocation) C compartments, but not from leaves, wood and roots. In the model proposed by Murty and McMurtrie

(a)

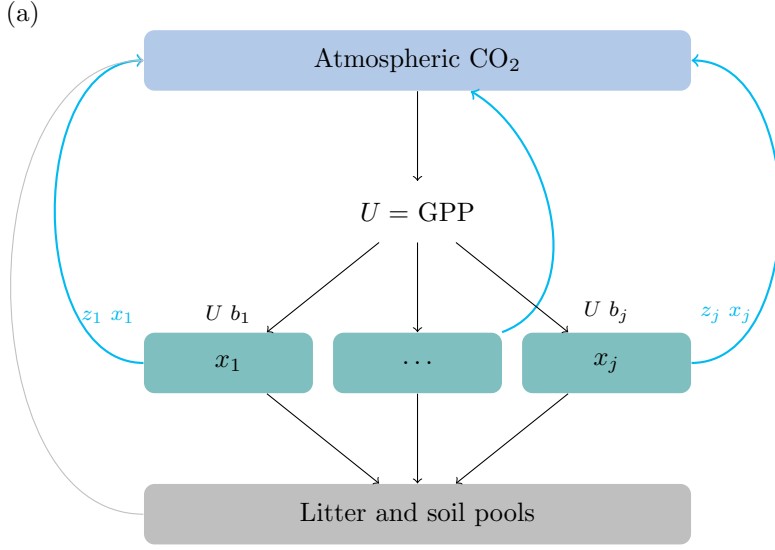

(b)

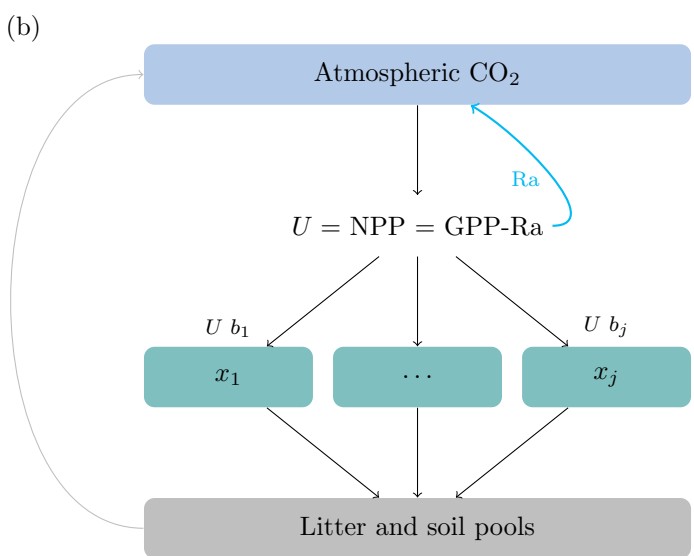

**Figure 1.** Conceptual diagram representing the difference between allocation schemes. (a) The source of carbon for allocation is GPP, split among the different vegetation pools $(x_1, \ldots, x_j)$ according to partitioning coefficients $(b_1, \ldots, b_j)$. The source of carbon for autotrophic respiration is the stock in the vegetation pools and it is computed according to the release coefficients $z_1, \ldots, z_j$ (see text for the definition of symbols). (b) The source of carbon allocation is NPP. In this case the functions used to obtain Ra may depend on the stock of carbon or nitrogen in vegetation pools, but Ra is subtracted from GPP before NPP is allocated. The carbon used for Ra never enters the vegetation pools and does not spend any time there.

(2000) there are different maintenance respiration terms that are subtracted from GPP before allocation, only respiration from the sapwood pool depends on its C stock, while other respiration terms depend on the N stock. In FOREST-BGC (Running and Coughlan, 1988), growth respiration and available C are calculated yearly, while maintenance respiration is calculated daily from the C stocks, but both respiration variables are subtracted from GPP. Similarly in CABLE (Wang et al., 2010, 2012), both maintenance and growth respiration are subtracted from GPP before allocating C from NPP.

The other eight models do not consider an explicit calculation of stock-dependent maintenance respiration, and also allocate carbon from NPP. Some of these models explicitly express that given the linear relationship between C canopy respiration and canopy photosynthesis, the autotrophic respiration is a fixed fraction of the total photosynthetic fixation. Some models that fall into this category are G'DAY (Comins and McMurtrie, 1993), DALEC (Williams et al., 2005), CASA (Potter et al., 1993), and TECO (Luo et al., 2012). Other models, such as the one proposed by Hilbert and Reynolds (1991) calculate the net C gain by subtracting dark respiration from GPP. Three other models do not mention respiration at all, and just partition C from a "rate of biomass production": CEVSA2 (Gu et al., 2010), the model proposed by King (1993), and the model proposed by DeAngelis et al. (2012) whose net carbon production depends on leaf C.

In many models, GPP and Ra occur at short timescales (half-hourly, hourly, or daily), computing the net carbon gain as an annual integral. Carbon allocation occurs at annual intervals, when the assimilated carbon that is not respired is assigned to a particular vegetation compartment. Therefore, the carbon that is respired at an intra-annual timescale never enters the vegetation pools.

The important point that we want to highlight here is that even though some models compute maintenance respiration based on knowledge of the carbon stock that needs to be maintained, this respiration is actually subtracted from GPP to obtain the net carbon gain that is subsequently allocated. Only in a few models, maintenance respiration is subtracted from a carbon stock such as a labile pool or other vegetation compartment, but most models can be written in the form of equation (1) with $U(t) = \mathrm{NPP}(t)$.

## 2.3 Continuous- versus discrete-time implementations

In addition to the issue of the source of carbon (GPP or NPP) used for allocation, there is a related problem in computing the age of Ra that emerges in model implementations that are discrete in time. Models based on ordinary differential equations such as those expressed as in equation (1) treat time as a continuous variable, but many models are implemented in discrete time steps where the carbon stocks of the previous time step are updated based on the functions defined by the model.

For example, in the Allometrically Constrained Growth and Carbon Allocation model (ACGCA, Ogle and Pacala (2009)) maintenance respiration is released from a transient non-structural carbon (NSC) pool. The carbon there is used for allocation to labile pools, structural tissue in the tree organs, and for respiration. It is a transient pool because the carbon is used immediately, which allows freshly assimilated carbon to be used for maintenance respiration. There are no issues with this implementation in continuous-time (Herrera-Ramírez et al., 2020), but in discrete-time implementations, at a one-year time-step in particular, a large portion of the carbon from the transient pool never enters the tree. The net carbon balance is still correct, but the model does not describe accurately the temporal dynamics of the carbon in the transient pool.

To compute maintenance respiration in this model, carbon can be used immediately and hence never enters the tree. Growth respiration on the other hand, happens at the same time step as carbon is allocated to the tree organs but with a one year time lag, one time step after it entered the transient pool from photosynthesis. Practically, this means that growth respiration happens one year later than maintenance respiration, and that carbon respired by maintenance has an age of zero. This age of respired carbon is not realistic when compared with measurements, which can be obtained at finer temporal resolutions and over a broader range.

## 3 Age of respired carbon obtained as the transit time distribution from models

The age of respired carbon can be obtained from ecosystem models, but the model structure and the form in which the source of carbon for allocation is represented has an impact on the age of carbon respired from ecosystems. Although most models do not represent carbon age explicitly, it can be computed using different computational approaches.

The age of respired carbon from ecosystem is characterized by its transit time distribution (Bolin and Rodhe, 1973; Thompson and Randerson, 1999; Sierra et al., 2021b). These distributions can be obtained from ecosystem carbon models using impulse response functions (Thompson and Randerson, 1999), a simulation approach that consists of applying a pulse of carbon to a model at equilibrium, where carbon stocks do not change over time, and then observing the respiration flux after the pulse. These distributions can also be obtained using the analytical formulas developed by Metzler and Sierra (2018) for models at equilibrium, or the approach described in Metzler et al. (2018) for models out of equilibrium.

The transit time distribution represents the proportions of respired carbon that have different ages, and it is usually a continuous function that results from a mixture of exponential functions (Metzler and Sierra, 2018). They can be obtained from any ecosystem model expressed in compartmental form as[1]

$$\frac{\mathrm{d}\boldsymbol{x}}{\mathrm{d}t} = \boldsymbol{u}(t) + \mathbf{B}(t)\,\boldsymbol{x} \tag{2}$$

where $\boldsymbol{u}(t)$ is a vector of carbon inputs to the system. In the framework of Luo et al. (2017, 2022), $\boldsymbol{u}(t) = U(t)\,\boldsymbol{b}$. The matrix $\mathbf{B}$ is a compartmental matrix with diagonal elements the cycling rates within the pools, and off-diagonal elements the transfer rates of carbon among the different pools. In the framework of Luo et al. (2017, 2022), $\mathbf{B}(t) = \xi(t)\mathbf{A}\,\mathbf{K}$. Respiration from each compartment $j$ can be obtained as the product of the amount of mass present in the system and a rate of release $z_j$,

$$r_j = z_j\,x_j. \tag{3}$$

This rate of release $z$ can be obtained from the compartmental matrix $\mathbf{B}$ as the negative sum of the entries of each column. It represents the fraction of carbon that leaves each pool and is not transferred to any other pool.

---

[1]For simplicity of notation, we use here the mathematical representation for linear autonomous systems, but the same arguments can be demonstrated for non-linear non-autonomous systems. However, the notation would be more complex to express and without additional insights.

The transit time distribution of carbon can be defined as the age of the respired carbon from the pools, and can be expressed as (Metzler and Sierra, 2018)

$$f_T(\tau) = \frac{1}{\|\boldsymbol{r}\|} \sum_j r_j f_{aj}(\tau) = \frac{1}{\|\boldsymbol{r}\|} \sum_j z_j \, x_j f_{aj}(\tau), \tag{4}$$

where $f_{aj}(\tau)$ is the age distribution function for pool $j$ as a function of the variable $\tau$ which represents age. The norm symbol $\|\cdot\|$ represents the sum of all elements of the vector. Equation 4 can be interpreted as the relative contribution of pools of different ages to the total respiration flux, in which each pool contains a mix of carbon of different ages characterized by a pool age distribution $f_a$.

If carbon is allocated from GPP, i.e. $\boldsymbol{u}(t) = \mathrm{GPP}(t) \, \boldsymbol{b}(t)$, autotrophic respiration can only occur directly from the carbon stored in the pools, and $z_j > 0$ for all pools where carbon is respired (Figure 1a). Conversely, if carbon is allocated from NPP, i.e. $\boldsymbol{u}(t) = (\mathrm{GPP}(t) - \mathrm{Ra}(t)) \, \boldsymbol{b}(t)$, no respiration occurs from vegetation pools and $z_j = 0$ for those pools (Figure 1b). We can infer then from equation (4) that for those pools that do not respire carbon ($z_j = r_j = 0$), their contribution to the transit time distribution is equal to zero.

For illustration purposes, we will show here predictions from the global carbon model developed by Emanuel et al. (1981) and used by Thompson and Randerson (1999) to represent differences between carbon allocation from GPP versus allocation from NPP. We will refer to these two cases as GPP-based versus NPP-based carbon allocation schemes. We use the model of Emanuel et al. (1981) because of its simplicity, which allows us to study differences in model structure without additional complexity.

At equilibrium, the GPP-based version of the model shows a continuous distribution of carbon that decreases with transit time (Figure 2). A large proportion of carbon is respired very quickly after photosynthetic fixation and smaller quantities are respired later on. In contrast, the NPP-based version of the model predicts that all autotrophically respired carbon has an age of zero, and respiration in later years comes only from heterotrophic pools. The median age of the respired carbon (50% quantile of the transit time distribution) in the GPP-based version of the model is 2.3 yr, i.e. 50 % of respired carbon is respired in less than 2.3 years. In contrast, in the NPP-based version of the model the median transit time is 0 yr, because the autotrophic respiration flux, which corresponds to 50 % of GPP, is removed immediately after photosynthetic fixation.

The GPP-based version of the model predicts a continuum of ages of respired carbon both for autotrophic and heterotrophic respiration (Figure 3). Although a large portion of autotrophic respiration is very young ($< 1$ year), a significant proportion is older and can be respired years after photosynthetic fixation.

## 4 Age of respired carbon obtained from radiocarbon measurements

Several studies have used radiocarbon-based methods to estimate the age of respired carbon form different compartments in ecosystems (e.g., foliage, wood, roots, and soil) (Carbone and Trumbore, 2007; Carbone et al., 2007, 2013; Muhr et al., 2013, 2018; Trumbore et al., 2015). In vegetation compartments, studies have focused mostly on individual trees rather than on a larger sample of trees within a forest stand. For healthy mature trees, small differences have been found between com-

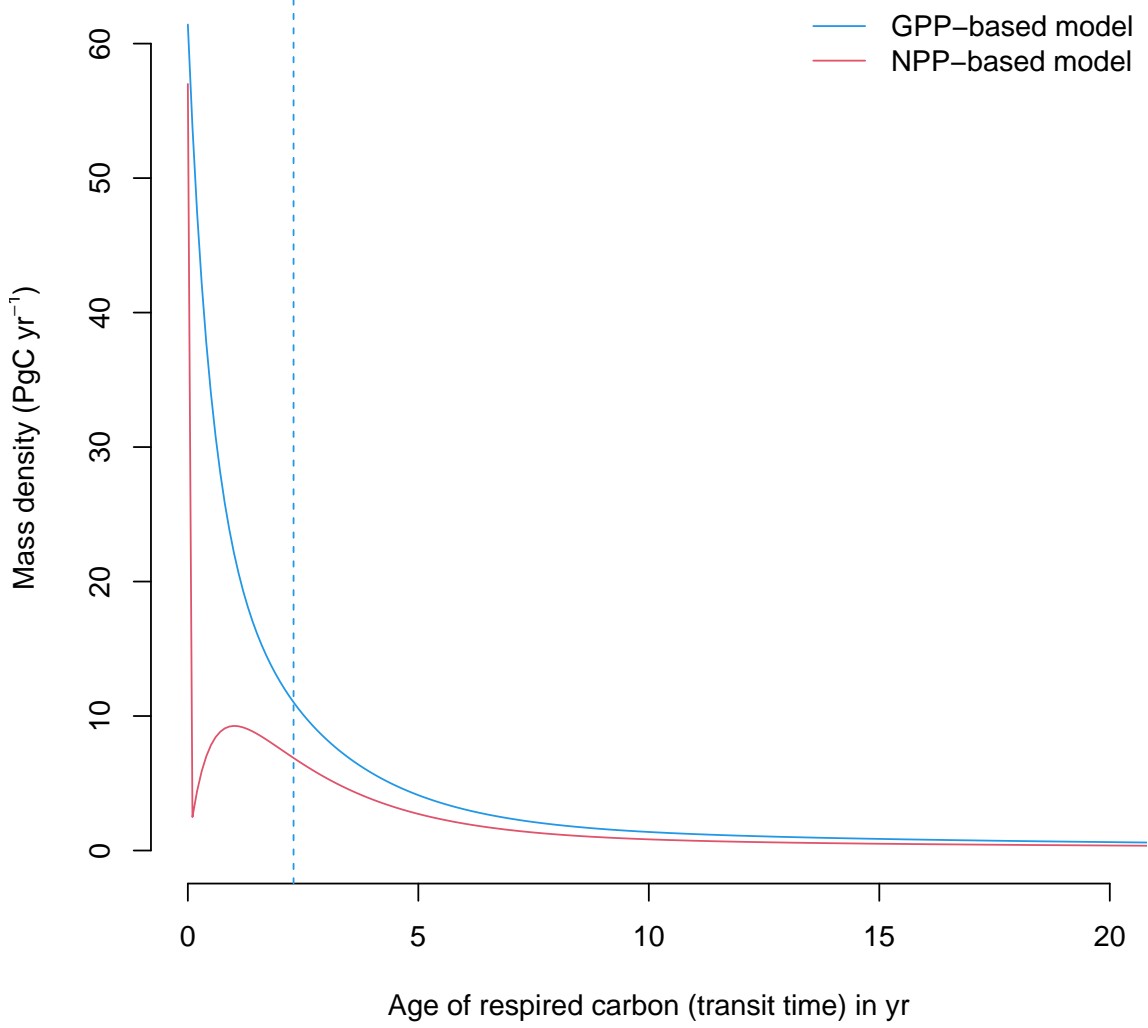

**Figure 2.** Transit time distributions obtained from the GPP- and the NPP-based versions of the model of Emanuel et al. (1981). The vertical dashed line represents the median transit time of the GPP-based model, which is 2.3 yr. For the NPP-version, the median transit time is 0 yr.

partments, for example carbon respired from leaves may be less than one year old (Carbone and Trumbore, 2007), while in roots and stems the respired carbon is on average older than one year, with a mix of carbon from recent assimilates and some contributions of old carbon from storage reserves (Muhr et al., 2018). There is empirical evidence that shows that the age of the respired carbon by trees can change during different seasons, and increases as trees are exposed to stress and have to use

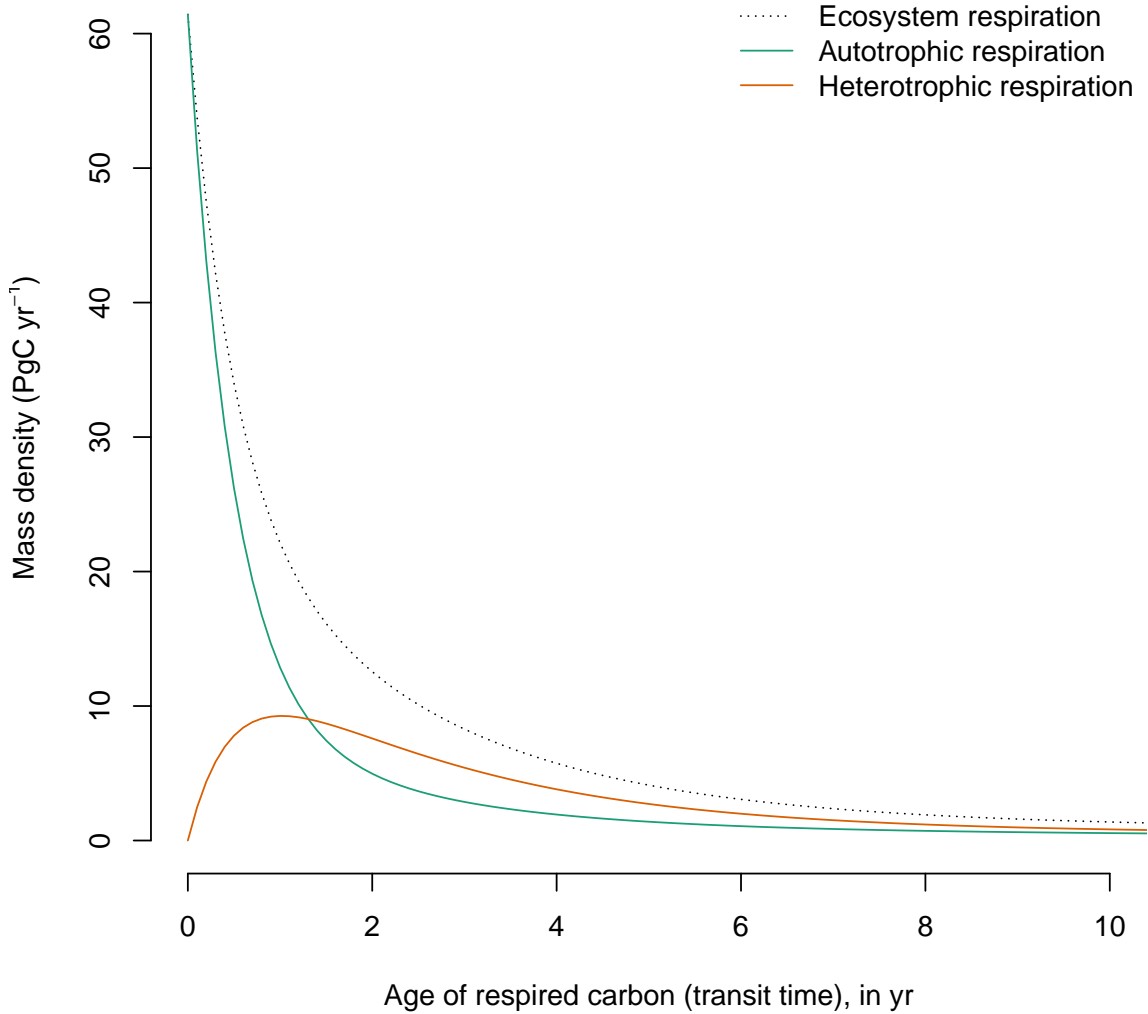

**Figure 3.** Contribution of autotrophic and heterotrophic respiration to the transit time distribution in the GPP-based version of the model of Emanuel et al. (1981). The age distribution of total ecosystem respiration is equivalent to the transit time distribution of the ecosystem.

their storage reserves to support metabolic activity. For instance, Carbone et al. (2013) reported ages of the respired $CO_2$ by the stem of *Acer rubrum* trees of 1.5 and $\sim$0 yr during spring and late summer, respectively. Muhr et al. (2013) reported ages of 2.5 and 3.3 yr for $CO_2$ respired from the stem of *Simaruba amara* trees during the dry and the wet season, respectively; 2 years old $CO_2$ from the stem of *Tachigali paniculata*; and 4.5 and 4 yr old $CO_2$ from stems of *Hymenolobium pulcherrimum*.

Herrera et al (in prep) found similar values as in these previous studies, 5 and 3 years old for $CO_2$ respired by in-stem samples of *Dacriodes microcarpa*, and 2.5 and 5 years old for $CO_2$ from *Ocotea leucoxylon* during the dry and wet season, respectively.
Some studies have also reported several years old respired $CO_2$, ranging from 1 to 5 yr from roots. Most of these studies report mean values of 4 yr old respired carbon from roots (Czimczik et al., 2006; Schuur and Trumbore, 2006; Carbone and Trumbore, 2007), but younger $CO_2$ (0.6 yr old) has been also reported by Hilman et al. (2021).

Physical damage such as girdling increases the age of the respired $CO_2$. For example, Muhr et al. (2018) reported 1 year old $CO_2$ respired by healthy *Scleronema micranthum* trees and 14 years old $CO_2$ respired by trees after one year of girdling. Also, Hilman et al. (2021) reported increases in the age of the respired carbon from roots, ranging from 0.4 yr from not girdled trees to 1.2 yr for tress after three months of girdling.

With very few exceptions, most of the empirical evidence supports the idea that respired carbon from vegetation parts is on average older than 1 yr, but higher values can be observed depending on the season or on whether trees suffer some form of physiological stress that decreases the supply of recent carbohydrates (Herrera-Ramírez et al., 2020).

This empirical evidence, which shows that the age of respired carbon spans from one to several years (Figure 4), is inconsistent with predictions from models in which carbon allocation is based on NPP where the age of respired carbon is exactly equal to zero (Figure 2).

## 5    Implications

The modeling choice of allocating carbon from NPP and not from GPP has important consequences for: (1) use of radiocarbon as an empirical constraint in model-data assimilation studies; (2) computing the transit time distribution of carbon in ecosystems; and (3) determining isotopic exchange between terrestrial ecosystems and the atmosphere. We briefly elaborate on these three implications in the following paragraphs.

First, as radiocarbon measurements become increasingly available for plant parts and respired $CO_2$ from ecosystems, there is an excellent opportunity to use these data for constraining vegetation models and testing model-based hypotheses. Model-data assimilation techniques are very powerful to reduce model structural uncertainty, and can be used to improve carbon allocation and respiration routines in models. However, as we have shown here, the age of respired $CO_2$ in NPP-based models is predicted as exactly zero, inconsistent with radiocarbon measurements. Therefore, by construction, NPP-based allocation schemes cannot be used to assimilate radiocarbon measurements and constrain allocation and respiration functions.

Second, the transit time distribution of carbon is an important metric to integrate many ecosystem-level processes and study ecosystem dynamics (Bolin and Rodhe, 1973; Thompson and Randerson, 1999; Sierra et al., 2017). Under the assumption of equilibrium, mean transit times of carbon in ecosystems can be obtained by dividing the total carbon stock over the total input flux. However, this approach provides no information on its underlying probability distribution. As shown above, the median transit time can deviate strongly from the mean, and the possibility to compute entire transit time distributions provides very useful information to integrate processes occurring at very different timescales (Sierra et al., 2021b). Models that subtract autotrophic respiration from GPP before allocating to plant parts cannot be used to compute entire transit time distributions,

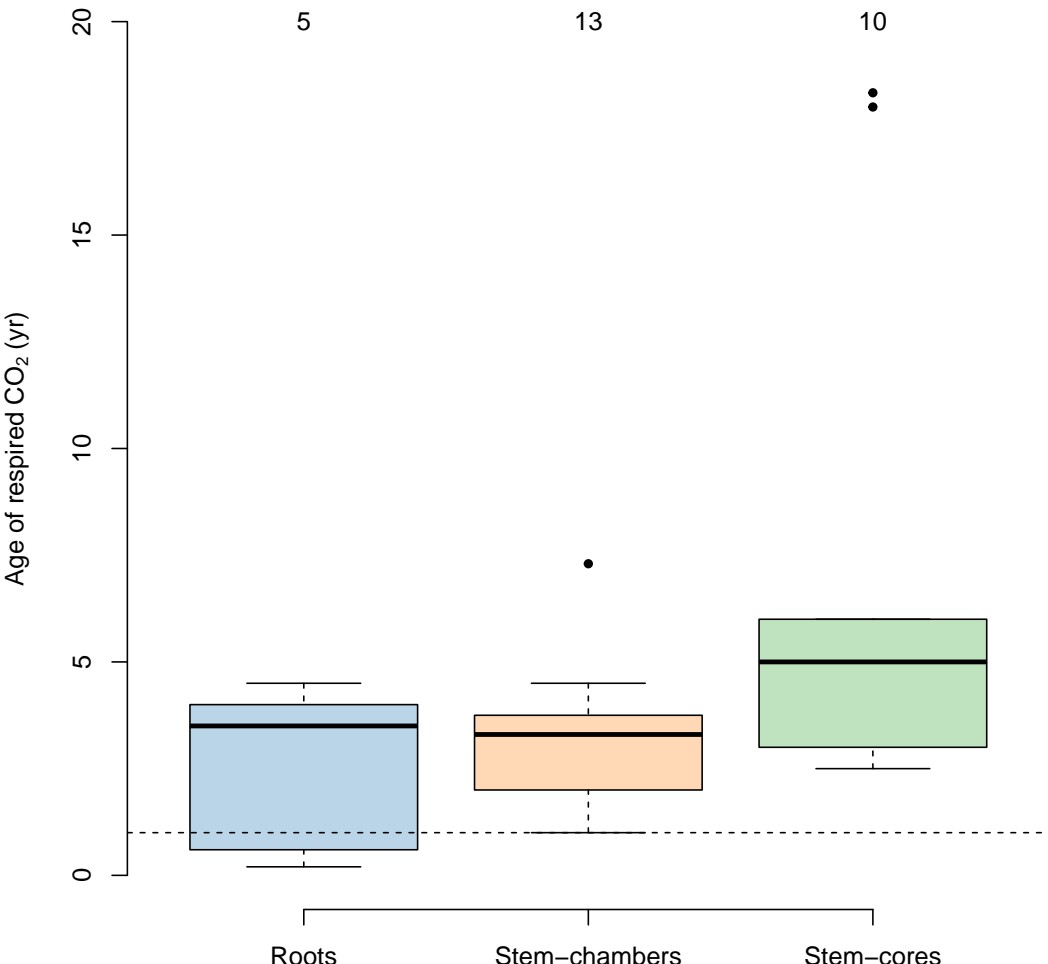

**Figure 4.** Age of C in respired $CO_2$ from roots and stems for different tree species from temperate and tropical forests obtained from radiocarbon measurements. Data for roots include both fine and coarse roots, and data for stems is split between chamber based measurements and incubations of tree cores. Numbers on top of the boxes represent the number of observations available to draw the boxplots. Values below the horizontal dashed line represent measurements of carbon younger than 1 yr.

missing on an opportunity to improve our understanding of the timescales of carbon exchange between ecosystems and the atmosphere.

Third, the choice of allocation scheme also has consequences for predicting the isotopic exchange of carbon between ecosystems and the atmosphere. For instance, predictions of radiocarbon signatures of respired $CO_2$ from the terrestrial biosphere show a large difference between the GPP- and NPP-based versions of the simple model (Figure 5). Because carbon spends

less time in NPP-based allocation schemes, the isotopic exchange between plant parts and the atmosphere occurs more rapidly than in the GPP-based representations. These differences may have important implications for predicting the isotopic disequilibrium between carbon reservoirs at the Earth system level (Randerson et al., 2002; Levin et al., 2021; Frischknecht et al., 2022). In a recent study, Frischknecht et al. (2022) reported that radiocarbon is exchanged too fast in the vegetation component of CLM5.0, inconsistent with previous reconstructions on the incorporation of radiocarbon in the terrestrial biosphere. A potential explanation for the inconsistencies identified by Frischknecht et al. (2022) may be the return of radiocarbon in Ra to the atmosphere immediately after GPP due to its allocation scheme.

## 6  Summary and recommendations

We have shown that models in which carbon allocation occurs after autotrophic respiration is subtracted from GPP (i.e. NPP-based models) predict that the age of respired carbon from vegetation pools is zero. This prediction contradicts empirical evidence based on the isotopic signature of respired $CO_2$ from plant parts, and suggests that GPP-based allocation schemes are more appropriate to represent carbon allocation and respiration in models. Models in which allocation is based on NPP miss on the opportunity to use radiocarbon data for constraining model parameters and improve their representation of vegetation processes. They are also unable to produce realistic transit time distributions of carbon, and can provide misleading predictions of isotopic exchange between ecosystems and the atmosphere.

We recommend modeling teams to revise the functions used to compute autotrophic respiration in models, in particular allowing carbon to enter into vegetation pools and then subtracting the autotrophic respiration flux from the standing carbon stock. The addition of a non-structural carbohydrate (NSC) pool can help to improve the dynamics of active carbon that is used to maintain metabolic processes (Ogle and Pacala, 2009; Ceballos-Núñez et al., 2018; Herrera-Ramírez et al., 2020), but models must ensure that the respired carbon is removed from these NSC pools and not from GPP. Models with one or two NSC pools can predict age distributions of C that span years to decades (Trumbore et al., 2015; Ceballos-Núñez et al., 2018; Herrera-Ramírez et al., 2020), consistent with observed data on the radiocarbon of NSCs and of respired $CO_2$. Similar modeling approaches can be implemented in other models. Nevertheless, care must be taken in avoiding artifacts introduced by the time step of the model in discrete-time implementations that may introduce time lags in the use of carbon for respiration. Differences in the time-step of discrete processes (e.g. GPP computed half-hourly versus annual allocation) pose important challenges for developing GPP-based allocation schemes. Future research should focus on developing strategies to collect the carbon produced at fast time scales and allocating carbon at monthly or seasonal scales. Data on phenology (Richardson et al., 2009, 2018) and tree-ring formation (Giraldo et al., 2022) can provide interesting insights for developing new C allocation functions at higher temporal resolution.

Another potential challenge to implement GPP-based carbon allocation schemes may be the availability and quality of GPP data. Traditionally, measurements of NPP and its components have been used to parameterize C allocation schemes, but new allocation functions may need to rely more on GPP data, NSC stocks, and radiocarbon measurements, integrated through data assimilation methods. Eddy-covariance estimates of GPP (Beer et al., 2010), together with new data on sun-induced fluoresce

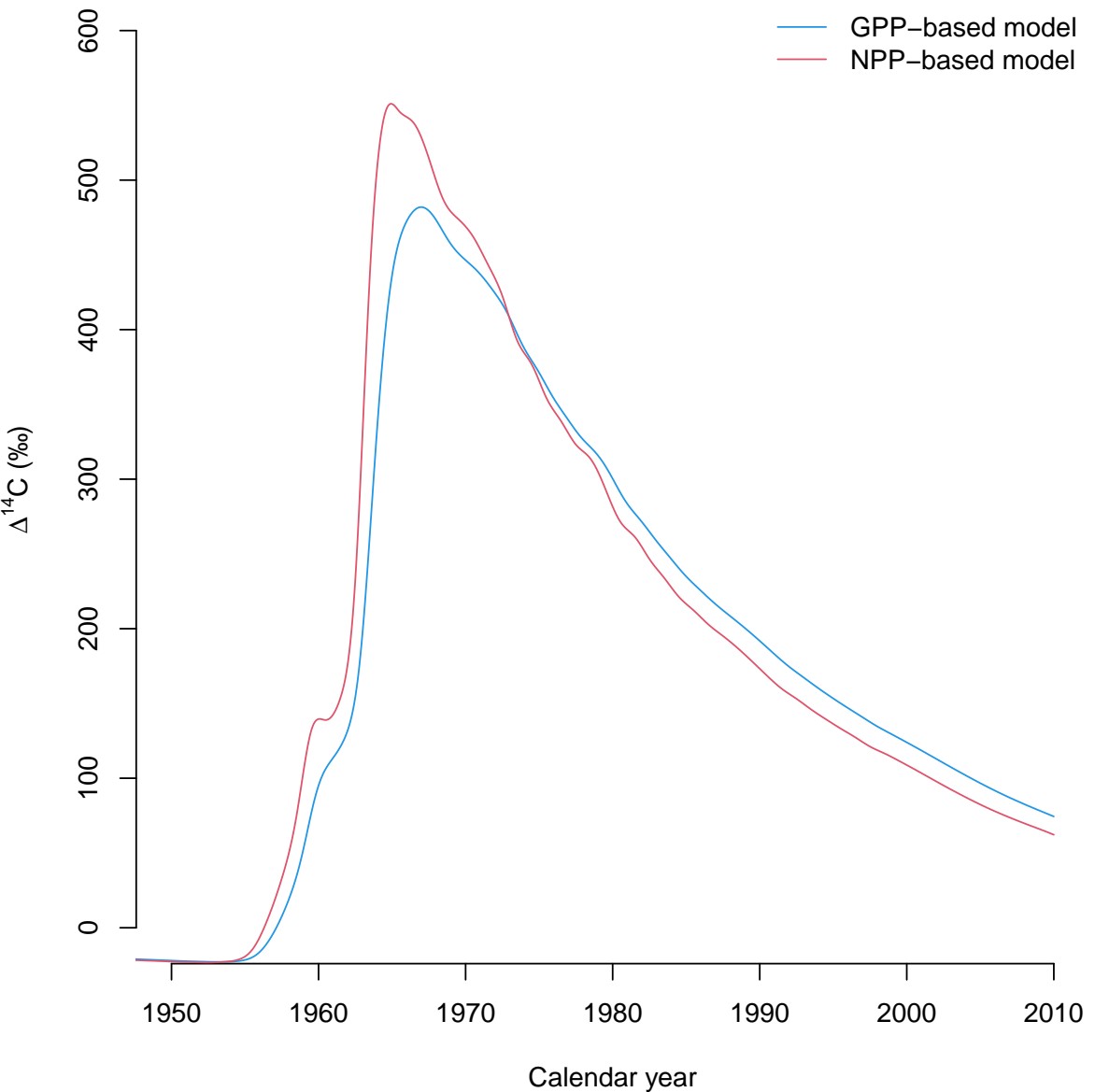

**Figure 5.** Radiocarbon in respired $CO_2$ (in $\Delta^{14}C$) predicted by the two versions of the simple model of Emanuel et al. (1981). The version in which carbon allocation occurs after Ra is subtracted from GPP (NPP-based model) predicts a faster exchange of radiocarbon with the atmosphere than the GPP-based version of the model where carbon stays for a longer time in the ecosystem.

(Gu et al., 2019) are providing now a wealth of data from a large number of ecosystems world wide. Synthesis efforts such as Fluxnet (Pastorello et al., 2020) and Fluxcom (Jung et al., 2020) provide well-curated data-products of global GPP. In

particular, Fluxcom combines remote sensing information with eddy-flux data to produce global gridded products of GPP at high spatial and temporal resolution, which could be of immense value for modeling studies.

Radiocarbon measurements in respired $CO_2$ from plant parts and whole ecosystem pools can also greatly help to test the mathematical structure of autotrophic respiration and allocation functions in models. These measurements are only available for a small set of sites, but future efforts should expand to more diverse ecosystems, capturing patterns induced by environmental drivers. Assimilation of radiocarbon data into ecosystem models offers large opportunities to improve our overall understanding of the timescales of carbon cycling in ecosystems and how they respond to environmental change.

*Code and data availability.* All data and code used for this manuscript is available in Zenodo at https://doi.org/10.5281/zenodo.6548611.

*Author contributions.* CAS designed research and wrote the manuscript. DHR compiled empirical studies on radiocarbon. VCN reviewed literature on models. HM analyzed differences between discrete and continuous implementations of Ra in models. HH wrote sections on physiology. All authors discussed ideas and contributed to writing.

*Competing interests.* The authors declare no competing interests.

*Acknowledgements.* Funding was provided by the German Research Foundation (SI 1953/2-2), the Max Planck Society, and the Swedish University for Agricultural Sciences. HM acknowledges the support of the Swedish Research Council for Sustainable Development FORMAS, under grant 2018-01820.

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
