# Peer review of "Ideas and perspectives: Allocation of carbon from Net Primary Production in models is inconsistent with observations of the age of respired carbon"

_EGUsphere, 2022_

## Referee Comment (RC2)

In this manuscript, Sierra *et al.* proposed that carbon allocation should be from GPP instead of NPP in ecosystem models since NPP-based models assumed autotrophic respiration only consumed fixed carbon immediately without transit time. They introduced the conceptual development of carbon allocation from NPP, reviewed 18 ecosystem models, and analyzed the distributions of carbon transit time between the two types of modeling schemes. The authors also showed that the NPP-based model conflicted with empirical evidence on plants' age of respired carbon. Overall, I enjoyed reading this paper. The logic of the manuscript is clear, and the presentation of the ideas and perspectives is precise and neat. I agree with the authors that there are many advantages to modeling carbon allocation from GPP. From a modeler's perspective, I found that some statements in the current version of the manuscript may need more discussion. From the perspective of an empirical ecologist, there are still some limitations to using GPP instead of NPP for carbon allocation. Please find my major and specific comments below.

Major comments:

1. GPP and carbon allocation are simulated with different time steps in many process-based ecosystem models. For example, GPP is commonly simulated during the daytime with a time step from half to three hours, but carbon allocation is updated daily. If the models adopted the GPP-based carbon allocation scheme, they have to improve the representations of diurnal changes in many processes related to plant growth and carbon allocation. These improvements could dramatically increase the complexity of the canopy module in the model. Some additional discussions on this issue could help modelers better understand the challenge of adopting the GPP-based carbon allocation scheme.

2. Fig. 1 showed a constant Ra/GPP ratio in most CMIP Earth system models. This pattern has also been reported in terrestrial ecosystem models (e.g., https://doi.org/10.1002/2016JG003384). As mentioned by the authors, a critical question is how to improve the modeling of autotrophic respiration in the models. Unlike leaf photosynthesis, the Ra scheme varies greatly among current ecosystem models. For example, in the CLM4.5 model, the growth respiration (Rg) is calculated as a factor of the total carbon in new growth on a given timestep, based on construction costs for a range of woody and non-woody tissues. The maintenance respiration (Rm) in CLM4.5 is a temperature function based on a base rate of Rm. However, in the JULES model, Rm is simulated from a moisture and nitrogen function based on dark leaf respiration. Rg in JULES is further calculated as a fraction of the difference between GPP and Rm. The authors have reviewed Rm in different models in section 2.1. It would be better if they could provide some details of the modeling of Ra in some specific models.

3. I agree with the authors that ecosystem models need to incorporate the non-structural carbon (NSC) pool dynamics. Adding the NSC pool into the equation (1) or (2) could affect the solution of carbon transit time because it changed the pool-flux structure in fig. 2. I'd like to suggest the authors discuss whether and how adding the NSC pool can influence the distributions of carbon transit time. Also, if we have enough data to parameterize the age of the NSC pool in the models?

4. There are some benefits to using NPP-based carbon allocation, especially in global models. First, the NPP-based scheme consists of more measurable parameters than the GPP-based scheme. The increasing observations of plant traits can be helpful in constraining those parameters. This advantage could be important for those non-woody ecosystems, in which the carbon allocation can be approximated by the annual growth of different plant tissues. Second, because the GPP-based scheme may need to increase the complexity of the canopy process, the computation cost could increase dramatically for data assimilation. Third, GPP itself is unmeasurable, so that the GPP uncertainty could propagate to the carbon allocation.

5. I also agree with the authors that radiocarbon data is helpful for improving the model. However, the measurements of radiocarbon are expensive in many countries. Maybe some introductions or discussions of available radiocarbon data from the ISRaD database are helpful for the readers.

Minor comments:

(1) It is better to give basic information about the function $f_{aj}(\tau)$ as described in Metzler *et al.* (2018). Some new readers could be unfamiliar with the matrix equation and its solution.
(2) Fig.1: Please add a few sentences to briefly describe those Earth system models.
(3) P13, L253-254: This statement might be too strong.
(4) The word "model" is used in different ways in the main text, such as ecosystem model, ecosystem carbon model, coupled carbon-climate model, land-surface model, carbon allocation model, etc. It is better to reduce the diversity of model types in the text.